# LOSING LESS: A LOSS FOR DIFFERENTIALLY PRIVATE DEEP LEARNING

## ABSTRACT

Differentially Private Stochastic Gradient Descent, DP-SGD, is the canonical approach to training deep neural networks with guarantees of Differential Privacy (DP). However, the modifications DP-SGD introduces to vanilla gradient descent negatively impact the accuracy of deep neural networks. In this paper, we are the first to observe that some of this performance can be recovered when training with a loss tailored to DP-SGD; we challenge cross-entropy as the de facto loss for deep learning with DP. Specifically, we introduce a loss combining three terms: the summed squared error, the focal loss, and a regularization penalty. The first term encourages learning with faster convergence. The second term emphasizes hard-to-learn examples in the later stages of training. Both are beneficial because the privacy cost of learning increases with every step of DP-SGD. The third term helps control the sensitivity of learning, decreasing the bias introduced by gradient clipping in DP-SGD. Using our loss function, we achieve new state-of-the-art tradeoffs between privacy and accuracy on MNIST, FashionMNIST, and CIFAR10. Most importantly, we improve the accuracy of DP-SGD on CIFAR10 by $4\%$ for a DP guarantee of $\varepsilon = 3$.

## 1 INTRODUCTION

Releasing machine learning (ML) models may risk the privacy of training data as ML models unintentionally leak information about the training data; this is due to limitations of learning, including overfitting (Song & Shmatikov, 2019) and data memorisation (Fredrikson et al., 2015; Carlini et al., 2019). The framework of Differential Privacy (DP) by Dwork et al. (2014) is the gold standard for formalising privacy guarantees. When training ML models, a DP training algorithm provides guarantees bounding the leakage of private data that may occur during training. This requires that one bound the influence of individual training examples on the output of the algorithm (e.g. parameters or gradients of the model). In addition to these strong theoretical guarantees, DP models are empirically resistant to various attacks – membership inference attacks (Shokri et al., 2017; Rahman et al., 2018), training data extraction (Carlini et al., 2019) and data poisoning attack (Ma et al., 2019).

Differentially Private Stochastic Gradient Descent, DP-SGD, trains a ML model under the framework of DP by (1) clipping per-example gradients to a fixed norm to bound their sensitivity and (2) adding noise to these clipped gradients before they are applied to update the model parameters (Abadi et al., 2016). When done separately and at the level of a minibatch, clipping (Menon et al., 2019; Zhang et al., 2019) or noising (Foret et al., 2021) effectively regularize learning because they respectively control the dynamics of iterates and smoothen the loss landscape. However, gradient clipping and noising are detrimental to model performance in DP-SGD because they are applied in combination and at the granularity of individual training examples. In addition to this, the accuracy of DP-SGD is negatively affected by the limited number of iterations that can be computed: each iteration of DP-SGD increases the privacy budget expended by learning.

Recent works have explored bounded activation functions (Papernot et al., 2021), publicly available feature extractors and data (Tramèr & Boneh, 2021), randomised smoothing (Wang et al., 2021) and gradient embedding (Yu et al., 2021) as a means to improve the tradeoffs between privacy and accuracy of DP-SGD. However, training deep neural networks with strong DP guarantees in DP-SGD still comes at a significant cost in accuracy for datasets like CIFAR10.

In this paper, we show that one key aspect of formulating the optimization problem solved in learning is left out of studies seeking to improve deep learning with DP: the loss function. All existing implementations of deep learning with DP we are aware of optimise cross-entropy (which performs well in the non-privacy-preserving setting with SGD). However, we first observe that optimising cross-entropy with DP-SGD leads to exploding model weights, layer pre-activations and logit values. This is true even if the activation functions themselves are bounded (Papernot et al., 2021). This phenomenon makes it difficult to control the sensitivity of the learning algorithm at a minimal impact to its correctness. Indeed, clipping and noising large gradients in DP-SGD introduces an information loss and additional biases. More precisely, the direction of a minibatch of clipped per-example gradients in DP-SGD is not necessarily aligned with the direction of the original gradients in SGD. Furthermore, we also note that the slow training convergence of cross-entropy (Allen-Zhu et al., 2019) negatively impacts the performance of DP-SGD for which a limited number of iterations are required to achieve strong privacy guarantees.

In this paper, to tackle these two limitations of the cross-entropy loss, we propose to tailor the loss function to the specifics of DP-SGD. We design a novel loss function that takes into account sensitivity, training convergence and the order in which examples are learned with the overarching goal of improving the tradeoffs between privacy and accuracy of DP-SGD. We achieve these objectives with a novel loss function that combines the Sum Squared Error (SSE) between the model logits and ground-truth labels, the focal loss (Lin et al., 2017), and a regularisation penalty on pre-activations.

The first two penalties impose a curriculum structure in the training to improve privacy-accuracy tradeoffs of DP-SGD by exploiting the generalisation and fast convergence speed (Wang et al., 2019; Guo et al., 2018) of curriculum learning (Bengio et al., 2009). We start training with SSE which has faster convergence (Allen-Zhu et al., 2019) and smaller gradients than cross-entropy. Then, we gradually shift toward emphasising hard-to-learn examples using the focal loss: this down-weights gradients assigned to examples that are already learned correctly to focus more on hard-to-learn and misclassified examples. Finally, our regularisation penalty prevents the weights from exploding to further reduce the magnitude of per-example gradients prior to clipping. Implementation-wise, our new loss function can be easily integrated within existing implementations of deep learning with DP; it requires a single line change to edit the loss function being optimized.

In summary, our main contributions are as follows:

- We are the first to investigate the loss function in the context of deep learning with DP. We analyse the effect of the loss function on the sensitivity of DP models by providing extensive analysis of the norm of activations, weights and gradients in Section 4.

- We propose a novel loss function tailored to deep learning with DP. In Section 3 and Section 4, we analytically and experimentally connect the superior performance of our proposed loss function to the gradient norm and direction, smoothness of the loss surface, and convergence of DP-SGD. Our proposed loss function better controls the sensitivity of the learning algorithm by better preconditioning gradients to being clipped and noised.

- We show in Section 4 that our proposed loss function is compatible with other DP-SGD improvement strategies, and more importantly allows us to establish new state-of-the-art tradeoffs between privacy and accuracy on several key benchmarks for deep learning with DP, including CIFAR10. Finally, we perform an ablation study to analyse the effect of each component of our proposed loss function on DP learning.

## 2   DIFFERENTIAL PRIVACY, DP-SGD AND STATE-OF-THE-ART APPROACHES

Trained machine learning models memorise and leak information about their training data (Shokri et al., 2017; Rahman et al., 2018; Song & Shmatikov, 2019; Fredrikson et al., 2015). Differential privacy (Dwork et al., 2014) is the established gold standard to reason about the privacy guarantees of learning algorithms. An algorithm is differentially private if its outputs are statistically indistinguishable on neighbouring datasets. More formally, a randomised learning algorithm $A$ is $(\varepsilon, \delta)$-differentially private if the following holds for any two neighbouring datasets $D$ and $D'$ that differ only in one record and all $S \in \text{Range}(A)$:

$$\Pr[A(D) \in S] \leq e^{\varepsilon}\Pr[A(D') \in S] + \delta. \tag{1}$$

The privacy budget $\varepsilon$ upper bounds the privacy leakage in the worst possible case: the smaller the $\varepsilon$, the tighter the upper bounds (or stronger privacy guarantee). Note that the factor $\delta$ is very small (generally it is chosen to be in the order of the inverse of the dataset size).

In order to train ML models with DP, we can add randomness to the learning algorithm in three ways: output perturbation (Wu et al., 2017; Zhang et al., 2017), objective perturbation (Chaudhuri et al., 2011; Iyengar et al., 2019) or gradient perturbation (Bassily et al., 2014; Abadi et al., 2016). Among these approaches, differentially private stochastic gradient, DP-SGD of Abadi et al. (2016), has established itself as the de facto strategy because of its versatility. DP-SGD perturbs gradients computed at each step of stochastic gradient descent using the Gaussian mechanism to achieve competitive tradeoffs between privacy and accuracy (Yu et al., 2020; Tramèr & Boneh, 2021). Next, we describe the process of training a classifier using DP-SGD on a private dataset.

Let $\mathcal{X} = \{\mathbf{x}_i | i = 1, ..., N\}$ be the training set containing $N$ private examples $\mathbf{x}_i$, and $\mathcal{Y} = \{\mathbf{y}_i | i = 1, ..., N\}$ be the ground-truth label set where each $\mathbf{y}_i$ is a $D$-dimension one-hot encoded vector. We consider a $D$-class classifier containing $M$ layers parameterised with weights $\mathcal{W} = \{\mathbf{W}^m | m = 1, ..., M\}$.

In each DP-SGD iteration, similarly to the non-private SGD one, the classifier receives each training example $\mathbf{x}_i$ from a minibatch containing $L$ data points that are sampled randomly from $\mathcal{X}$. The activation of each intermediate layer $m$ of size $d_m$, $\mathbf{a}_i^m = f(\mathbf{h}_i^m) \in \mathbb{R}^{1 \times d_m}$, is computed by applying a non-linear function, $f(\cdot)$, on the pre-activation $\mathbf{h}_i^m = \mathbf{W}^m \mathbf{a}_i^{m-1}$. The pre-activation is a linear combination of the weight of that layer, $\mathbf{W}^m$, and the activation of the previous layer, $\mathbf{a}_i^{m-1}$. The last layer outputs the logit values associated with each class without any activation function as $\mathbf{a}_i^M = \mathbf{h}_i^M = \mathbf{W}^M \mathbf{h}_i^{M-1} \in \mathbb{R}^{1 \times D}$. The cross-entropy loss, $\mathcal{L}_{\text{CE}}$, for each training example $\mathbf{x}_i$ is computed as

$$\mathcal{L}_{\text{CE}} = -\sum_{d=1}^{D} y_{(i,d)} \log(p_{(i,d)}), \quad \text{where} \quad p_{(i,d)} = \frac{e^{a_{(i,d)}^M}}{\sum_{d'=1}^{D} e^{a_{(i,d')}^M}}. \tag{2}$$

The probability that $\mathbf{x}_i$ belongs to $d$-th class, $p_{(i,d)} \in [0, 1]$, is computed by applying a Softmax function on the logit value of its corresponding class $a_{(i,d)}^M$.

The DP-SGD optimiser computes *per-example* gradient of $\mathcal{L}_{\text{CE}}$ with respect to $\mathcal{W}$, as opposed to per minibatch gradient in the SGD optimiser, to bound the influence of each individual training example on the output of the learning algorithm (i.e. weight gradients). The gradients, $\mathbf{G}^m$, for each training example $\mathbf{x}_i$ are computed from the last layer and back-propagated until the first layer as:

$$\mathbf{G}_i^M = (\mathbf{p}_i - \mathbf{y}_i)^T \mathbf{a}_i^{M-1}, \quad \mathbf{G}_i^{M-1} = \left((\mathbf{p}_i - \mathbf{y}_i)\mathbf{W}^M\right)^T \mathbf{a}_i^{M-2},$$
$$\mathbf{G}_i^{M-2} = f'\left((\mathbf{p}_i - \mathbf{y}_i)\mathbf{W}^M \mathbf{W}^{M-1}\right)^T \mathbf{a}_i^{M-3}. \tag{3}$$

One can continue this gradient computation until the first layer. All the per-example gradients are concatenated as $\mathbf{G}_i = [\mathbf{G}_i^1; ...; \mathbf{G}_i^M]$, where ; denotes the concatenation operation. As there is no a priori bound on $\mathbf{G}_i$, the $l_2$ norm, $\| \cdot \|_2$, of each $\mathbf{G}_i$ are artificially clipped by $C$ to bound the influence of each $\mathbf{x}_i$ on the final gradients, $\mathbf{G}$:

$$\mathbf{G} = \frac{1}{L}\left(\sum_{i=1}^{L} \mathbf{G}_i + \mathcal{N}(0, \sigma^2 C^2 \mathbf{I})\right) \quad \text{where} \quad \mathbf{G}_i \leftarrow \mathbf{G}_i \cdot \min(1, \frac{C}{\|\mathbf{G}_i\|_2}). \tag{4}$$

Finally, the classifier weights are updated through the average of the clipped and noisy (scaled by the noise multiplier $\sigma$) per-example gradients of a minibatch.

The accuracy of classifiers trained with DP-SGD is lower than that of classifiers trained by non-private SGD. Recently, researchers investigated the choice of $f(\cdot)$ and data representation for improving the accuracy of the classifier trained by DP-SGD. Papernot et al. (2021) demonstrated that exploiting a bounded family of activation functions instead of the more commonly employed unbounded ReLU activation for the choice of $f(\cdot)$ can decrease the bias introduced by DP-SGD and improve the tradeoffs between privacy and accuracy of DP-SGD. Their approach though does not fully bridge the gap between non-private and private learning for datasets like CIFAR10. Tramèr & Boneh (2021) proposed to train the classifier on a representation of data outputted by a public ScatterNet feature extractor as opposed to using the raw pixels $\mathbf{x}_i$. This however requires access to public data in addition to the private dataset.

## 3 PROPOSED APPROACH

We propose to design a new loss function tailored to DP-SGD. Our goal is to converge faster with smaller gradient norms, control better the sensitivity of the learning algorithm, increase tolerance to noise, and effectively learn both easy and hard examples. Indeed, these address several crucial differences between DP-SGD and its non-private counterpart SGD:

- Due to per-example gradient clipping (recall Equation 4), information contained in gradients whose magnitude is too large is discarded (see Figure 1.c). This cannot be compensated by tuning the training algorithm, e.g., by increasing the model's learning rate. This is because clipping is done at the granularity of individual training examples. Together with the noise injected (recall Equation 4), this biases learning and implies that DP-SGD takes a different optimisation path compared to SGD.
- As shown in Equation 3, the model weights[1] contribute to the computation of all gradients. In practice, this leads to model weights exploding in DP-SGD. Therefore, one way to decrease the magnitude of gradients is to prevent the model weights from exploding, especially the weight of the last layer which contributes in the gradient of all the layers.
- The number of training iterations are limited in DP-SGD because each iteration increases the risk of privacy leakage. Hence, faster convergence is beneficial to DP-SGD as well as ensuring both easy and hard examples are attended to during the limited training run.

Next, we describe our choice of loss function. We analyse how our new loss function improves the tradeoffs between privacy and accuracy of DP-SGD. We uncover several effects: our loss limits the information loss of gradient clipping, speeds up convergence, recovers a generalisation boost comparable to the one achieved by batch normalisation, and improves learning's tolerance to noise.

### 3.1 OUR LOSS

We propose to learn the easy examples at the beginning of the training using the per-example Sum Squared Error (SSE) between the logit values and one-hot vector labels:

$$\mathcal{L}_{\text{SSE}} = \frac{1}{2}\|\mathbf{h}_i^M - \mathbf{y}_i\|^2, \quad \text{where} \quad \|\mathbf{h}_i^M - \mathbf{y}_i\|^2 = \sum_{d=1}^{D}(h_{(i,d)}^M - y_{(i,d)})^2. \tag{5}$$

Later in training, we exploit the focal loss $\mathcal{L}_{\text{Focal}}$ to learn hard-to-learn examples. $\mathcal{L}_{\text{Focal}}$ modifies the cross-entropy loss to reduce the loss of easy, well-classified examples, letting the model focus more on hard, misclassified examples. $\mathcal{L}_{\text{Focal}}$ multiplies a factor $(1 - p_{(i,t)})^\gamma$ based on the probability of the ground-truth class $p_{(i,t)}$ to the cross-entropy of each training example $\mathbf{x}_i$ as:

$$\mathcal{L}_{\text{Focal}} = -(1 - p_{(i,t)})^\gamma \sum_{d=1}^{D} y_{(i,d)} \log(p_{(i,d)}), \tag{6}$$

where the tunable focusing parameter $\gamma$ adjusts the down-weighting rate: the higher the $\gamma$, the higher the down-weight rate of easy, well-classified examples. Note that the focal loss is equivalent to cross-entropy loss, when $\gamma = 0$.

To further decrease the gradient magnitudes and prevent the explosion of intermediate weights, we impose a regularisation penalty on the intermediate pre-activations as:

$$\mathcal{L}_{\text{Reg}} = \sum_{m=1}^{M-1} \frac{1}{d_m}\|\mathbf{h}_i^m\|^2. \tag{7}$$

Finally, our proposed loss function $\mathcal{L}$ combines $\mathcal{L}_{\text{Focal}}$, $\mathcal{L}_{\text{SSE}}$ and $\mathcal{L}_{\text{Reg}}$ as:

$$\mathcal{L} = \alpha\mathcal{L}_{\text{Focal}} + (1 - \alpha)\mathcal{L}_{\text{SSE}} + \frac{(1 - \alpha)}{\beta}\mathcal{L}_{\text{Reg}}, \tag{8}$$

where we set hyper-parameter $\alpha = \text{Sigmoid}(e_c - e_t)$ (current epoch, $e_c$, and threshold epoch, $e_t$) to enable curriculum learning where easy examples are learned in the early training iterations using $\mathcal{L}_{\text{SSE}}$ which eases learning of hard examples in the later training iterations using $\mathcal{L}_{\text{Focal}}$. We perform a hyper-parameter search to set the best values for $\alpha$, $\beta$ and $\gamma$.

---

[1]Note that all the terms except those corresponding to weights are bounded in Equation 3.

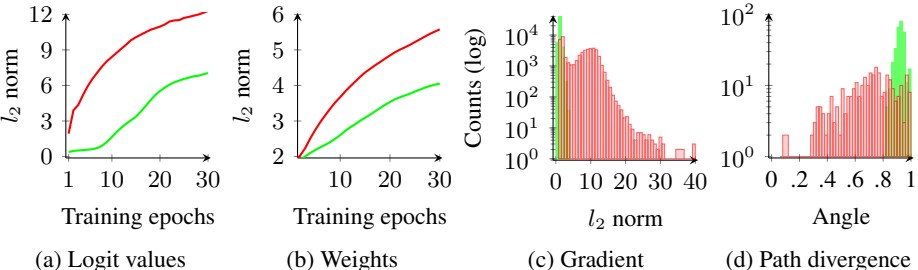

(a) Logit values     (b) Weights     (c) Gradient     (d) Path divergence

Figure 1: $l_2$ norm of the logit values (a), $l_2$ norm of the last layer weights (b), histogram of per-sample gradient magnitude in the first epoch (c), and divergence histogram of optimisation path in aggregated clipped gradients from the optimisation path in unclipped ones (d) when we train the classifier on CIFAR10 using cross-entropy loss or our proposed loss function. Minimising cross entropy loss using DP-SGD leads to logit exploding (a) and weight exploding (b). However, our proposed loss function prevents logit exploding (a) and weight exploding (b). Therefore, per-sample gradient magnitudes (c) obtained based on our proposed loss function is smaller than per-sample gradient magnitudes obtained based on cross entropy loss. In addition to this, the per-sample gradients of our proposed loss function is much more condensed. Finally, direction of the gradients (d) of our proposed loss function is more aligned with the unclipped gradients.

### 3.2 AN ANALYSIS OF OUR LOSS

**Our loss limits information loss from clipping.** Figure 1 shows that under the supervision of $\mathcal{L}$, we are able to control the sensitivity of the learning algorithm. In particular, $\mathcal{L}_{\text{Reg}}$ on the intermediate pre-activations and $\mathcal{L}_{\text{SSE}}$ on the logit values both prevent logit values and weights from exploding, which decreases the magnitude and variance of per-example gradients (see Equation 3). Smaller and more condensed gradients decrease the negative impact of gradient clipping on the trajectory of the gradient descent as shown in Figure 1.d.

**Our loss yields faster convergence.** $\mathcal{L}_{\text{SSE}}$ improves the tradeoffs between privacy and accuracy of DP-SGD with strong privacy budgets as $\mathcal{L}_{\text{SSE}}$ converges exponentially $O(e^{-t})$, where $t$ is the steps, to 0 loss, while cross-entropy loss only converges at $O(1/t)$ as shown in Allen-Zhu et al. (2019).

**Our loss achieves an effect similar to batch normalisation.** The magnitude of gradients computed when optimising $\mathcal{L}_{\text{SSE}}$ with DP-SGD is smaller than for gradients computed when optimising cross-entropy, as $\mathcal{L}_{\text{SSE}}$ is defined on logit values $\mathbf{h}_i^M$. This prevents logit values and consequently $\mathbf{W}^M$ from exploding (per-example logit values are computed based on the weight of the last layer). Controlling $\mathbf{W}^M$ can limit the changes of weights in other layers as their gradients are a function of $\mathbf{W}^M$ and per layer gradient function is locally lipschitz. Preventing logits from exploding may also help recover the generalisation boost of batch normalisation (Dauphin & Cubuk, 2021). Indeed, batch normalisation cannot be used in differentially private learning because the technique shares information across different training examples contained in a single minibatch. This violates the privacy analysis of DP-SGD. Therefore, our proposed $\mathcal{L}_{\text{SSE}}$ penalty may also be beneficial to learning with DP-SGD because it achieves a similar effect to batch normalisation (Dauphin & Cubuk, 2021).

**Our loss improves tolerance to noise.** Wang et al. (2021) showed that smoother loss functions improve the generalisation bounds and accuracy of DP learning by making the loss surface more tolerant to noise. The most common smoothness notion is based on the Lipschitz constant of the gradient of a function.

**Definition 1 (Smoothness (Nesterov, 2003))** *A loss function $\mathcal{L}$ is $\beta$-smooth if*
$$\|\nabla\mathcal{L}_w - \nabla\mathcal{L}_{w'}\|_2 \leq \beta\|w - w'\|_2,$$

where $\nabla\mathcal{L}_w$ is the gradient of the loss function with respect to the weights and $\beta$ is the smoothness constant. Smaller $\beta$ indicates a smoother function.

We now derive a lemma illustrating how our loss also favours smoothness of the loss being optimised by DP-SGD.

**Lemma 1** *When singular values are bounded by $\sigma_{max}$, we have for all $w$ and $w'$ in a simply connected set that $\|\nabla\mathcal{L}_w - \nabla\mathcal{L}_{w'}\|_2 \leq \sigma_{max}\|w - w'\|_2$.*

Proof.
*Let $H_w$ be the hessian of the loss function with respect to the weights $w$.*

$$\|\nabla\mathcal{L}_w - \nabla\mathcal{L}_{w'}\|_2 = \|(\int_0^1 H_{w+t(w-w')} \cdot (w-w')dt)\|_2 \leq \int_0^1 \|H_{w+t(w-w')} \cdot (w-w')\|_2 dt$$

$$\leq \int_0^1 \sigma_{max}\|w-w'\|_2 dt \leq \sigma_{max}\|w-w'\|_2,$$

(9)

*where $\sigma_{max}$ is the maximum of the first singular value along the line from $w$ to $w'$.*

Therefore, the existence of local bounds on the singular values results in local bounds on the smoothness. We empirically observe in Figure 2 that the singular values of our loss function are smaller than cross-entropy loss function. This is particularly true at the beginning of training. This, thus, suggests that our proposed loss function is smoother and more noise-tolerant than cross-entropy.

## 4 VALIDATION

We validate the benefit of deploying our proposed loss function in improving how DP-SGD tradeoffs privacy and accuracy. To do so, we show our loss further improves the state-of-the-art results of the two approaches for deep learning with DP described in Section 2–1) `CE-T-CNN` (Papernot et al., 2021): using the cross-entropy loss function together with an end-to-end CNN classifier equipped with Tanh activation functions; 2) `CE-T-PFE_CNN` (Tramèr & Boneh, 2021): using the cross-entropy loss function, this time combining the ScatterNet public feature extractor (PFE) with a private CNN classifier whose internal activations are also Tanh. To ensure a fair evaluation, we use the same datasets, MNIST, FashionMNIST and CIFAR10, and the same classifiers (described in Appendix A) as these two baseline ap-

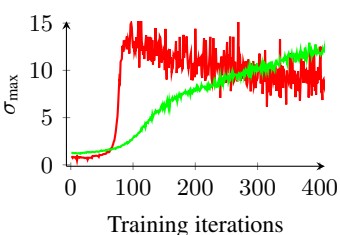

Figure 2: Analysing the smoothness of the landscape of our proposed loss function and cross-entropy on MNIST. Our loss function obtains smaller $\sigma_{max}$ (the maximum of the first singular value), making it more smooth than cross entropy.

proaches. In addition to this, we fix the DP-SGD configuration across all approaches to the best hyperparameter values reported in the hyperparameter search done by Tramèr & Boneh (2021) (see Appendix B). Note that our results would only be improved by additional hyperparameter search on the DP-SGD configuration. For the hyperparameters introduced by our own loss function, we use a search on the values as described in detail in Appendix B. We note that none of the prior work we compare against captures the privacy cost of the hyperparameter search itself when reporting DP guarantees achieved. The cost of this search should however be moderate, as analysed by Liu & Talwar (2019).

### 4.1 IMPROVING THE TRADEOFFS BETWEEN PRIVACY AND ACCURACY

Figure 3 and 4 compare the privacy-accuracy tradeoffs achieved by our proposed loss function and prior approaches for DP training. Coloured regions represent the range of model accuracy with respect to the privacy budget $\varepsilon$ needed to achieve this accuracy. Experiments are repeated 5 times; the minimum, median and maximum of the 5 experiments are each highlighted by a line.

When learning without privacy, end-to-end classifiers achieve a test accuracy on CIFAR10, FashionMNIST and MNIST of 76.6%, 89.4% and 99.0%, respectively. Instead, when learning with DP-SGD, the maximum test accuracy (across 5 runs) of our first baseline `CE-T-CNN` for $\varepsilon < 3$ are 59.8%, 86.9%, 98.2% on CIFAR10, FashionMNIST and MNIST, respectively. This illustrates how CIFAR10 is the most challenging dataset for DP-SGD training among the datasets we considered: there is about 20% gap in accuracy between the privacy-preserving and non-privacy-preserving settings. Our proposed loss function reduces this gap significantly, even more so for CIFAR10 than simpler datasets like FashionMNIST and MNIST. For example, training an end-to-end classifier using our loss function on CIFAR10 achieves 63.2% test accuracy, instead of 59.8% with the cross-entropy loss. While our second baseline `CE-T-PFE_CNN` sees stronger accuracy than the first

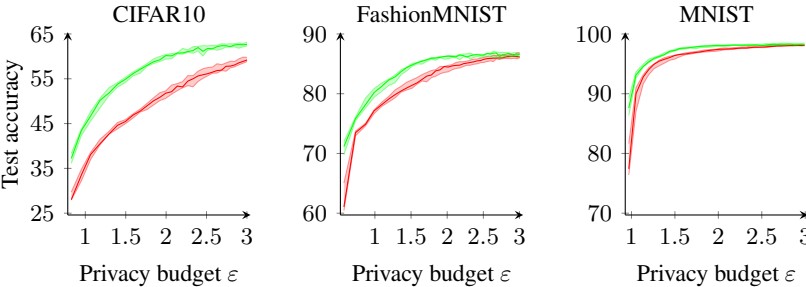

Figure 3: Deployment of our loss function in DP-SGD training of an end-to-end classifier for 5 runs. Comparison between the test accuracy of `CE-T-CNN` (Papernot et al., 2021) (——) and `O-T-CNN`, ours (——), as a function of privacy budget $\varepsilon$.

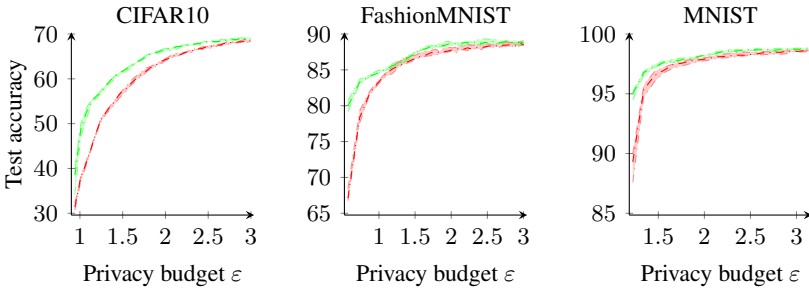

Figure 4: Deployment of our loss function in DP-SGD training of a public feature extractor followed by a classifier for 5 runs. Comparison between the test accuracy of `CE-T-PFE_CNN` (Tramèr & Boneh, 2021) (–·–·–) and `O-T-PFE_CNN`, ours (– – –), as a function of privacy budget $\varepsilon$.

baseline `CE-T-CNN` because the learner additionally has access to public data, our proposed loss function also improves its ability to tradeoff privacy and accuracy across the entire $\varepsilon$ regime.

The improvement of our proposed loss function to privacy-accuracy tradeoffs is strongest in the initial epochs. These epochs correspond to stronger privacy guarantees (i.e., smaller values of $\varepsilon$) because the privacy guarantee of each individual step of DP-SGD needs to be composed across the entire training run; hence each step further increases the privacy budget expended. For example, our proposed `O-T-CNN` trains an end-to-end classifier with $55.4\%$ test accuracy for $\varepsilon < 1.5$, compared to $45.9\%$ only with our first baseline `CE-T-CNN`. In another example, `O-T-PFE_CNN` improves the test accuracy of our second baseline `CE-T-PFE_CNN` from $64.5\%$ to $66.8\%$ for $\varepsilon < 2$.

## 4.2 ANALYSIS OF GRADIENTS COMPUTED BY DP-SGD FOR `O-T-CNN`

We first analyse the impact of the loss function on the model weights, pre-activations, and logit values. Figure 5 (first row) shows the impact our loss function has on **model weights**. We visualise the average $l_2$ norm of model weights at each epoch. Across the three datasets, optimising `CE-T-CNN` with DP-SGD leads to model weight explosion. This is not the case when learning with our proposed loss. In particular, note how `O-T-CNN` prevents last layer weights from exploding, which is important because they contribute to the gradients for all layers. Next, Figure 5 (second row) shows the impact of our loss function on **pre-activation and logit norms**. Although `CE-T-CNN` uses bounded tanh activation functions to prevent activations from exploding, the pre-activations and logits still explode because model weights are unbounded and explode themselves (as visualized above). Instead, replacing the cross-entropy loss function with our loss function decreases this phenomenon for both pre-activations and logits. This is explained by our pre-activation regulariser penalty but also the fact that the sum-squared-error penalty is defined over logits.

**Reducing the bias of gradient clipping.** We provide the histogram of per-example gradient $l_2$ norms in Figure 6. For better visualisation, we clip the x-axis between $[0, 100]$ and use a log scale for the y-axis. The gradient $l_2$ norms of `O-T-CNN` are smaller than their counterpart for `CE-T-CNN`. This is a consequence of our observations above: our proposed loss function prevents several terms including in the gradient computation from exploding. To further demonstrate the importance of better preconditioning gradients to the clipping performed by DP-SGD, we visualize the divergence

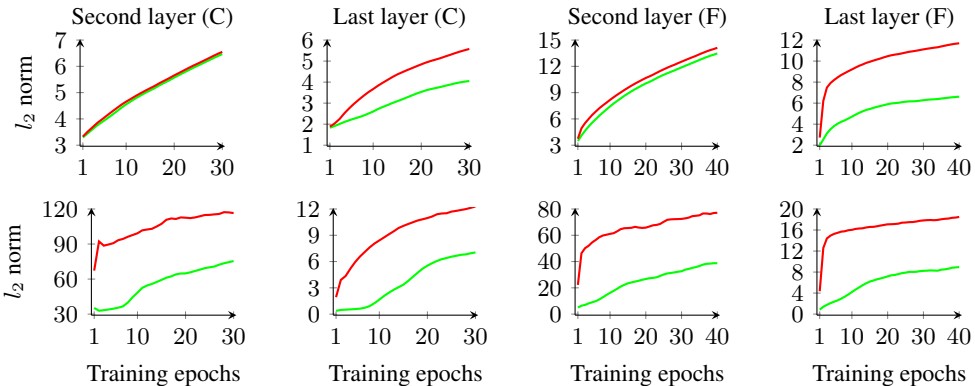

Figure 5: $l_2$ norm of weights (first row) and pre-activations (second row) in `CE-T-CNN` (——) and `O-T-CNN` (——) using CIFAR10 (C) and FashionMNIST (F). See Appendix C for results of MNIST and other layers.

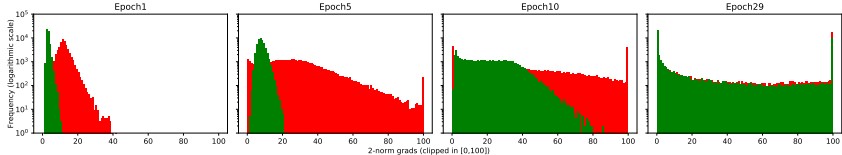

Figure 6: Histogram of $l_2$ norm of per-example gradient in `CE-T-CNN` and `O-T-CNN` on CIFAR10. Histogram for other epochs and other datasets are presented in Appendix E.

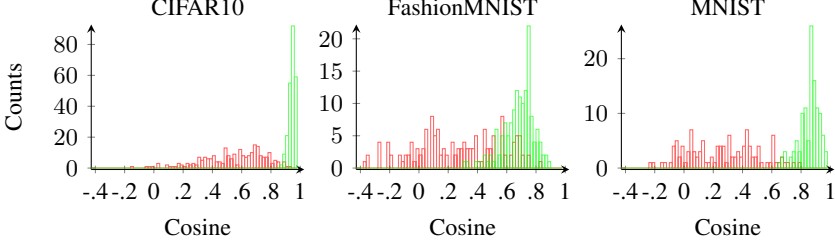

Figure 7: Cosine similarity between aggregated of clipped per-example gradients and aggregated of unclipped per-example gradients in first epoch of `CE-T-CNN` (——) and `O-T-CNN` (——).

between the path taken by aggregated *clipped* per-example gradients and aggregated *unclipped* per-example gradients in Figure 7. For each minibatch, we compute the cosine similarity between the average of per-example clipped gradients and the average of per-example unclipped gradients. The direction of clipped gradients in `O-T-CNN` is more aligned with the direction of their unclipped counterparts than is the case for the baseline approach. This suggests that our proposed loss function decreases the information loss and bias introduced by DP-SGD.

**Learning hard vs. easy examples.** Finally, we visualise the confusion matrices of per-class training accuracy for non-private learning with SGD, private learning with the `CE-T-CNN` baseline, and private learning with our approach `O-T-CNN` in Figure 8. Qualitatively speaking, the per-class training accuracy of `O-T-CNN` is closer to the non-private one than is the case for the baseline `CE-T-CNN`. During the initial epochs of learning, `O-T-CNN` converges faster especially on easy-to-learn examples thanks to the sum-squared-error loss function. For example, the training accuracy of `O-T-CNN` for "car" and "truck" are 51% and 48%, while `CE-T-CNN` only achieves 34% and 37%, respectively, on these classes. In addition to this, `O-T-CNN` deals better with hard-to-learn examples (e.g. "cat", "dear" or "dog" classes) in later epochs thanks to the focal loss. For example, the training accuracy of "cat" class is 49% at epoch 20 of `O-T-CNN`, but is only 35% for `CE-T-CNN`.

## 4.3 ABLATION STUDY

Next, we present an ablation study to tease out the contribution of each of the three components of our proposed loss when it comes to improving privacy-accuracy tradeoffs in deep learning with

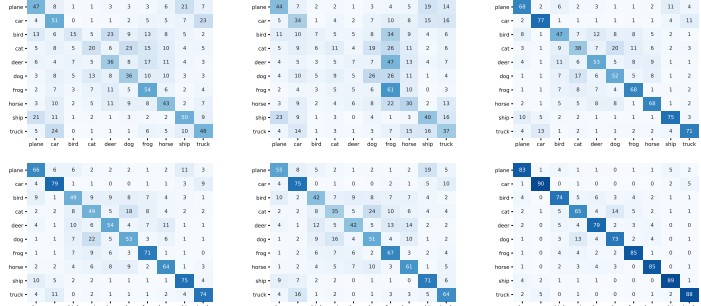

Figure 8: Confusion matrices that show per-class training accuracy of `O-T-CNN` (first column), `CE-T-CNN` (second columns) and non-private SGD approach (last columns). The first and second row show confusion matrices of CIFAR10 at epoch 2 and 20, respectively. See Appendix D for the results of other epochs as well as FashionMNIST and MNIST datasets.

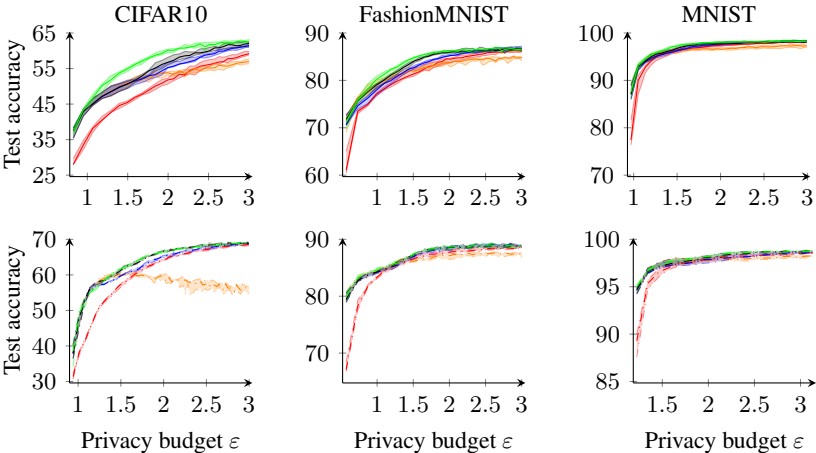

Figure 9: Ablation study of DP-SGD training of an end-to-end classifier (first row) and a public feature extractor followed by a classifier (second row) for 5 runs. Loss functions under study: CE, SSE, SSE+CE, SSE+Focal and SSE+Focal+Regulariser.

DP, namely sum-squared-error, focal loss and regularisation penalty. To do that we consider four combinations of these penalties: only using sum-squared-error, a combination of sum-squared-error and cross-entropy, a combination of sum-squared-error and focal loss, and finally our proposed loss function that combines all three-sum-squared-error, focal loss and regulariser. We do not consider other combinations because they are not meaningful. To ensure a fair evaluation, we perform a hyperparameter search for the combination of sum-squared-error and cross-entropy as well as the combination of sum-squared-error and focal loss, similarly to the search performed for our proposed loss function (see Appendix B). The privacy-accuracy tradeoffs of these four combinations is shown in Figure 9. While the sum-squared-error improves the privacy-accuracy tradeoff in initial epochs, the focal loss improves it for later epochs. Finally, imposing a regularisation penalty on pre-activations further improves these privacy-accuracy tradeoffs. Our proposed loss thus achieves the best possible performance across each of these settings and none of its penalties can be omitted.

## 5 CONCLUSION

In this paper, we showed that designing a loss function tailored to the specificities of DP-SGD, namely per-example gradient clipping and a limited number of iterations, significantly improves tradeoffs between privacy and accuracy. We reduced the bias introduced by gradient clipping using a pre-activation regularisation term. In addition to this, we improved the convergence speed by imposing a curriculum structure in learning with the sum-squared-error and focal loss. As future work, we will evaluate our loss function for DP-SGD in other domains such as text.

## 6 Reproducibility Statement

We submit our code as the supplementary material. Our code is based on the public Opacus library in PyTorch. We use the publicly available datasets. For the theoretical smoothness result, we provide clear explanations of lemma and complete proof.

## 7 Ethics Statement

Our work improves the tradeoffs between privacy and accuracy of training machine learning models in privacy-preserving setting. Therefore, we expect the broader impact of our work to be generally positive especially in privacy-sensitive applications such as health-care or language modeling where we work with sensitive datasets in practice.

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

Table 1: The architecture of end-to-end CNNs for Fashion-/MNIST (Papernot et al., 2021).

| Layer | Parameters |
|-------|-----------|
| Conv | 16 filters of 8x8, stride 2, padding 2 |
| MP | 2x2, stride 1 |
| Conv | 32 filters of 4x4, stride 2, padding 0 |
| MP | 2x2, stride 1 |
| FC | 32 units |
| FC | 10 units |

Table 2: The architecture of end-to-end CNNs for CIFAR10 (Papernot et al., 2021).

| Layer | Parameters |
|-------|-----------|
| Conv x2 | 32 filters of 3x3, stride 1, padding 1 |
| MP | 2x2, stride 2 |
| Conv x2 | 64 filters of 3x3, stride 1, padding 1 |
| MP | 2x2, stride 2 |
| Conv x2 | 128 filters of 3x3, stride 1, padding 1 |
| MP | 2x2, stride 2 |
| FC | 128 units |
| FC | 10 units |

Table 3: The architecture of CNNs trained on top of public ScatterNet feature extractor for Fashion-/MNIST (Tramèr & Boneh, 2021).

| Layer | Parameters |
|-------|-----------|
| Conv | 16 filters of 3x3, stride 2, padding 1 |
| MP | 2x2, stride 1 |
| Conv | 32 filters of 3x3, stride 1, padding 1 |
| MP | 2x2, stride 1 |
| FC | 32 units |
| FC | 10 units |

Table 4: The architecture of CNN trained on top of public ScatterNet feature extractor for CIFAR10 (Tramèr & Boneh, 2021).

| Layer | Parameters |
|-------|-----------|
| Conv x2 | 64 filters of 3x3, stride 1, padding 1 |
| MP | 2x2, stride 2 |
| Conv x2 | 64 filters of 3x3, stride 1, padding 1 |
| MP | 2x2, stride 2 |
| FC | 10 units |

## A  CLASSIFIERS

Table 1 and Table 2 show the architecture of end-to-end CNNs for MNIST, FashionMNIST and CIFAR10, suggested by Papernot et al. (2021). Table 3 and Table 4 show the architecture of CNNs trained on ScatterNet features for MNIST, FashionMNIST and CIFAR10, suggested by Tramèr & Boneh (2021).

## B  HYPERPARAMETERS

Table 5 reports hyperparameter values of the DP-SGD configuration across all approaches, suggested by Tramèr & Boneh (2021). Table 6 reports the hyperparameter values of our proposed loss function, obtained through a hyperparameter search on values reported in Table 7. However, we evaluate the performance of our loss function for each hyperparameter value using all three datasets in Figure 10. We can observe that the impact of hyperparameter values on the performance of our loss function is negligible. Therefore, the improvement of our loss function is not sensitive to the hyperparameter values.

Table 5: Training hyperparameter values obtained through line search Tramèr & Boneh (2021).

| Dataset | Learning rate | Momentum | Batch size | Epochs | Noise scale | Clipping norm |
|---------|--------------|----------|-----------|--------|-------------|---------------|
| End-to-end CNN | | | | | | |
| CIFAR10 | 1 | 0.9 | 1024 | 30 | 1.54 | 0.1 |
| Fashion-MNIST | 4 | 0.9 | 2048 | 40 | 2.15 | 0.1 |
| MNIST | 0.5 | 0.9 | 512 | 40 | 1.23 | 0.1 |
| CNN trained on ScatterNet features | | | | | | |
| CIFAR10 | 4 | 0.9 | 8192 | 60 | 5.67 | 0.1 |
| Fashion-MNIST | 4 | 0.9 | 2048 | 40 | 2.15 | 0.1 |
| MNIST | 1 | 0.9 | 1024 | 25 | 1.35 | 0.1 |

Table 6: The value of the hyperparameter of our loss function.

|              | CIFAR10                              | FashionMNIST                         | MNIST                                  |
|--------------|--------------------------------------|--------------------------------------|----------------------------------------|
| `O-T-CNN`    | $e_t = 7, \beta = 11, \gamma = 5$    | $e_t = 0, \beta = 1, \gamma = 5$     | $e_t = 0, \beta = 1, \gamma = 5$       |
| `O-T-PFE_CNN`| $e_t = 2, \beta = 1, \gamma = 2$     | $e_t = 2, \beta = 51, \gamma = 2$    | $e_t = 2, \beta = 11, \gamma = 10$     |

Table 7: Hyperparameters for our proposed loss function.

| Parameter | Value                              |
|-----------|------------------------------------|
| $e_t$     | Start=0, End=10, Step-size=1       |
| $\beta$   | Start=1, End=200, Step-size=10     |

## C $l_2$ NORM OF PRE-ACTIVATION AND WEIGHTS

Figure 11 and Figure 12 show the $l_2$ norm of weights and pre-activation of models trained by DP-SGD approaches on CIFAR10, FashionMNIST and MNIST.

## D CONFUSION MATRICES OF PER-CLASS ACCURACY

Figure 13 visualises the confusion matrices of per-class training accuracy for non-private learning with SGD, private learning with the `CE-T-CNN` baseline, and private learning with our approach `O-T-CNN` using CIFAR10, FashionMNIST and MNIST.

## E PER-EXAMPLE GRADIENTS

Figure 14 provides the histogram of per-example gradient $l_2$ norms in `CE-T-CNN` and `O-T-CNN` on CIFAR10, FashionMNIST and MNIST.

## F COMPARING SSE WITH SAE AND HUBER

In this section, we analyse the performance of SSE, Sum Absolute Error (SAE) and Huber-Loss in DP-SGD training.

The SAE is defined as:

$$\mathcal{L}_{\text{SAE}} = |\mathbf{h}_i^M - \mathbf{y}_i|, \quad \text{where} \quad |\mathbf{h}_i^M - \mathbf{y}_i| = \sum_{d=1}^{D} |(h_{(i,d)}^M - y_{(i,d)})|. \tag{10}$$

The $d$-th element of Huber loss is defined as:

$$\mathcal{L}_{\text{Huber}}(d) = \begin{cases} \frac{1}{2}(h_{(i,d)}^M - y_{(i,d)})^2/\beta & |(h_{(i,d)}^M - y_{(i,d)})| < \beta \\ |(h_{(i,d)}^M - y_{(i,d)})| - \frac{1}{2}\beta & \text{otherwise} \end{cases} \tag{11}$$

Figure 15 shows the tradeoffs between the privacy and accuracy of DP-SGD training using SSE, SAE and Huber loss. The performance of SSE is better than SAE. The performance of Huber loss increases as we increase $\beta$, activating SSE part and deactivating SAE part of Huber loss in Equation 11.

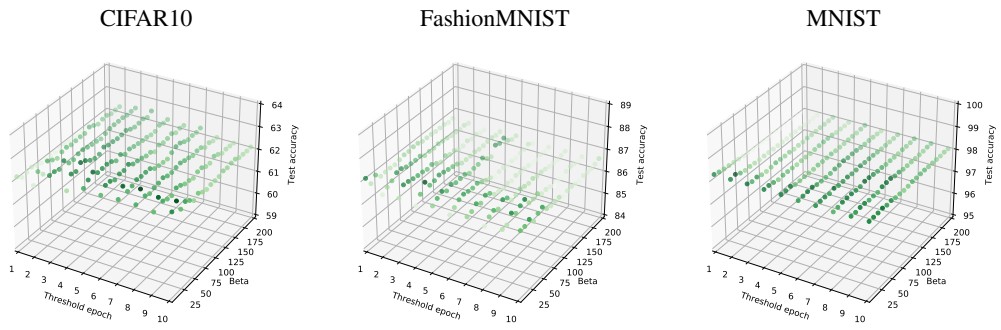

Figure 10: Test accuracy of `O-T-CNN` as a function of hyperparameter values of our loss function.

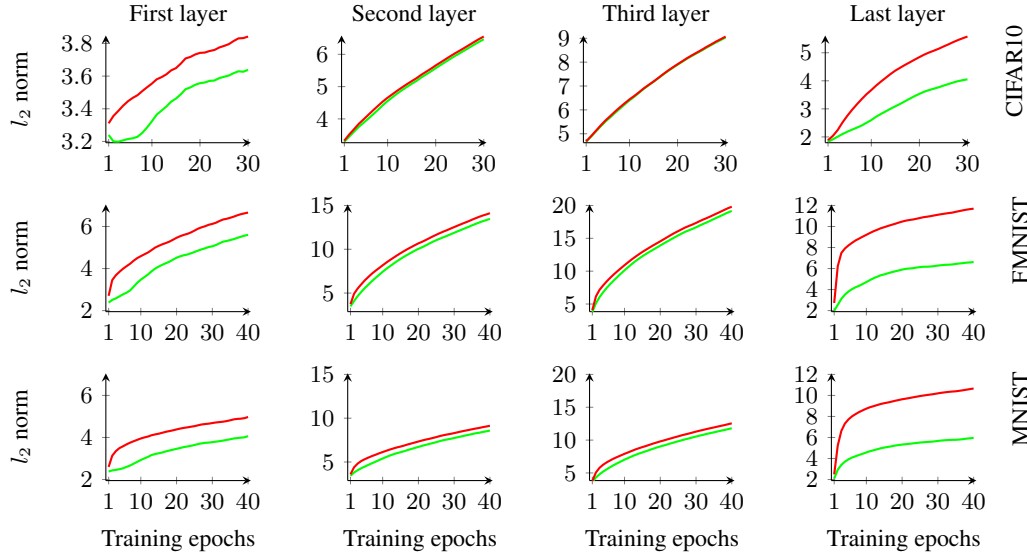

Figure 11: $l_2$ norm of model weights in `CE-T-CNN` (——) and `O-T-CNN` (——) using CIFAR10, FashionMNIST and MNIST.

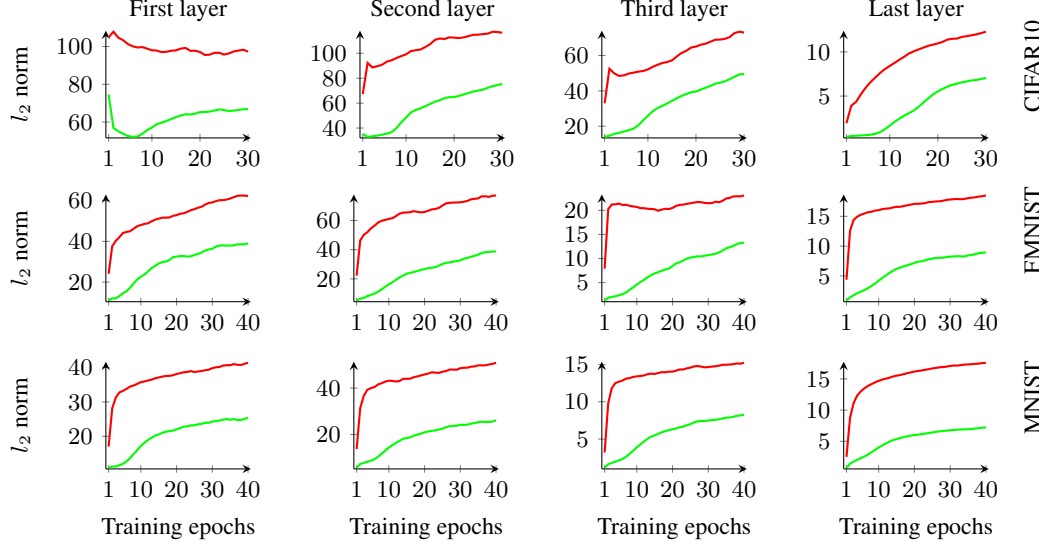

Figure 12: Pre-activation $l_2$ norm per layer of `CE-T-CNN` (——) and `O-T-CNN` (——) on CIFAR10, FashionMNIST and MNIST.

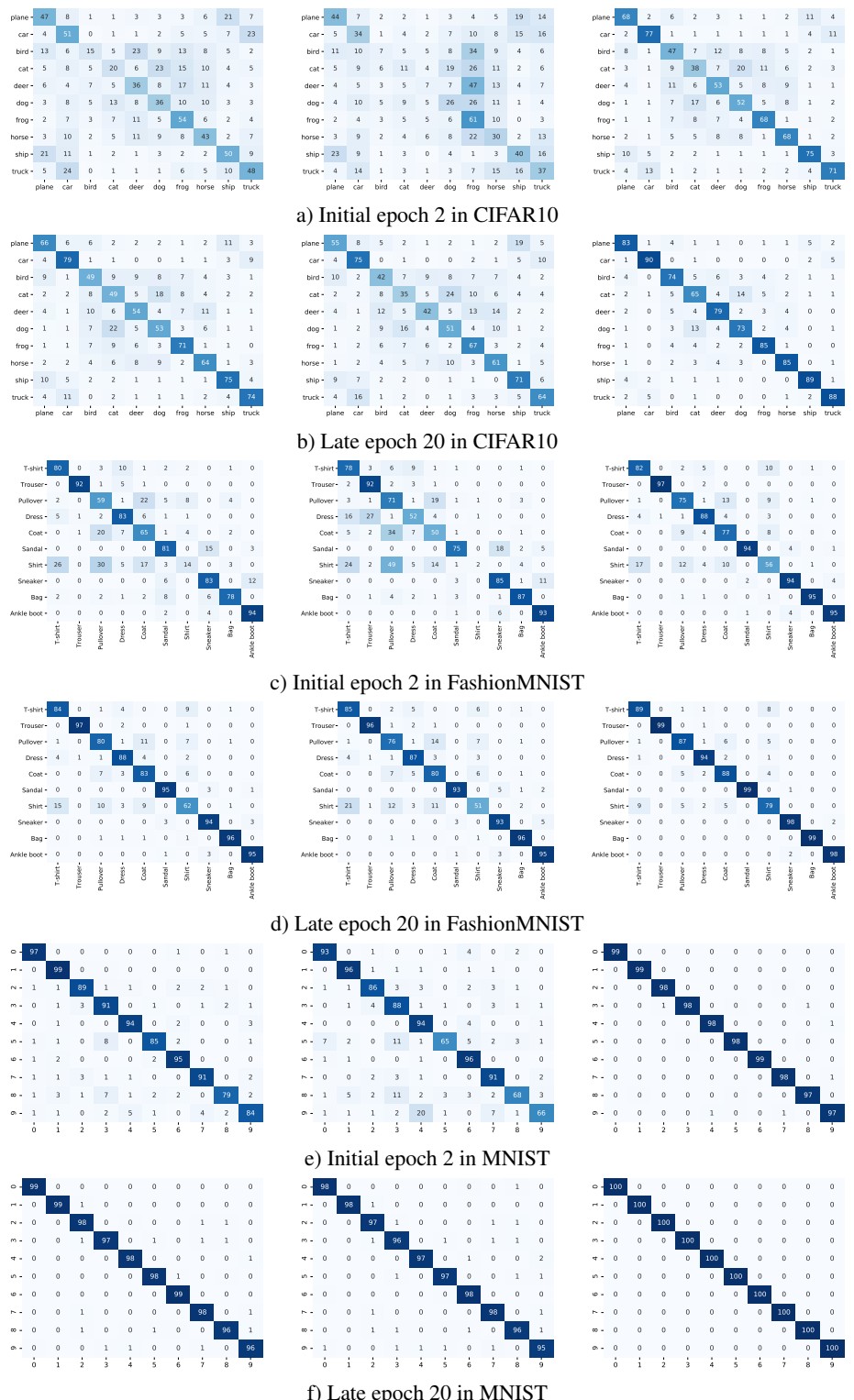

Figure 13: Confusion matrices that show per-class training accuracy of `O-T-CNN` (first column), `CE-T-CNN` (second columns) and non-private SGD approach (last columns) using CIFAR10, FashionMNIST and MNIST.

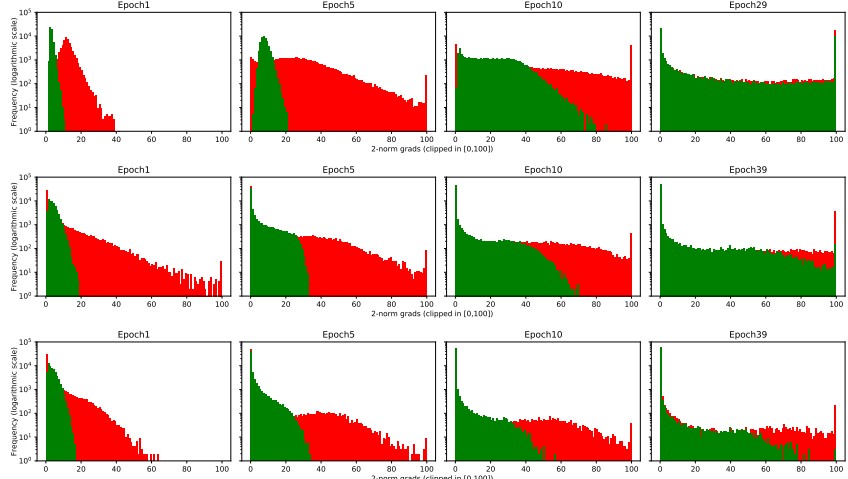

Figure 14: Histogram of $l_2$ norm of per-example gradient in `CE-T-CNN` and `O-T-CNN` on CIFAR10 (top row), FashionMNIST (middle row) and MNIST (bottom row).

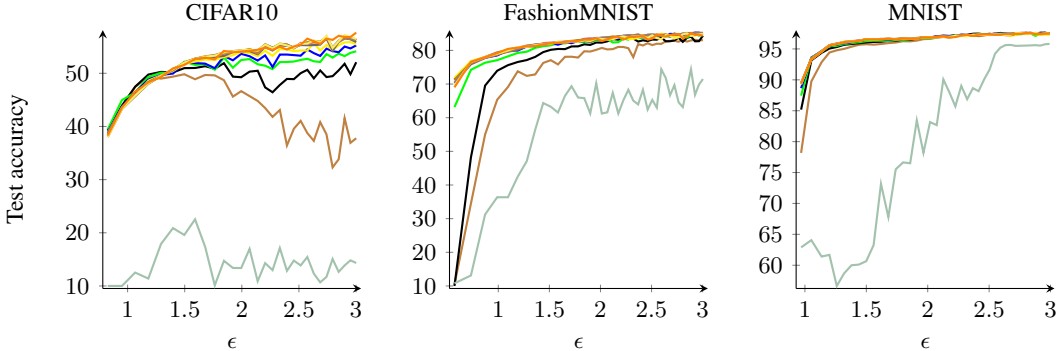

Figure 15: DP-SGD training of an end-to-end classifier using SSE ——, SAE —— and Huber loss ($\beta = .1$ ——, $\beta = .2$ ——, $\beta = .3$ ——, $\beta = .4$ ——, $\beta = .5$ ——, $\beta = .6$ ——, $\beta = .7$ ——, $\beta = .8$ ——, $\beta = .9$ —— and $\beta = 1$ ——).

