# OpenReview forum: "Losing Less: A Loss for Differentially Private Deep Learning"
_ICLR.cc/2022/Conference — ICLR 2022 Submitted_

### Official Review · Reviewer_jXWf · 2021-10-24

**Correctness:** 4
**Technical Novelty And Significance:** 2
**Empirical Novelty And Significance:** 2
**Recommendation:** 6
**Confidence:** 2

**Main Review:**

I think this paper is clear and easy-to-follow. The underlying philosophy that maybe DP learning should not use the same loss function as regular non-DP learning is interesting. This empirical paper has analyzed a set of computer vision datasets and the improvement does exists. I appreciate the authors efforts to do ablation study and to give details of network architectures etc. in the appendix.

However, my main concern is the message that this paper is trying to convey. First of all, the experiments are not comprehensive: only computer vision tasks are included and only on toy datasets. I believe including recommendation system and NLP datasets will make the statement of using their new loss much more convincing. Secondly, the empirical improvement is not significant. The seemly most significant improvement (as is the one presented in the abstract) is 4% on CIFAR10. Given that non-private model can easily get over 90% on CIFAR10, I believe the gap is not really closed by introducing the new loss. Lastly, the lack of theory may make people wonder, does this improvement really hold for general DP training? At most we can say on specific datasets and specific models, using the new loss is beneficial. Also all three components of the new loss already exist in regular training, so the current approach seems a simple combination without sufficient justification.

**Summary Of The Paper:**

This paper proposes a new loss that consists of three parts, namely MSE+focal loss+regularization loss. The new loss is supposed to benefit DP deep learning in comparison to the cross-entropy loss. Accuracy improvement has been observed on MNSIT, FashionMNIST and CIFAR10.

**Summary Of The Review:**

The paper is clear and easy-to-follow, combining three existing losses to form a new one that empirically improves DP accuracy on some vision datasets. However, the experiments should include non-vision tasks and provide (at least discuss) the insight from a theoretical viewpoint. Also the improvement is not significant enough for ICLR venue.

---

> ### Author Response · Authors · 2021-11-16
> **Authors’ reply to Reviewer jXWf**
>
> We thank the reviewer for the comments, finding our paper clear and considering the need of designing loss function for DP-SGD interesting.  In the text below we split comments and respond to them. The reviewer’s comments are in bold font, and the authors’ reply is in regular font.
>
> > **First of all, the experiments are not comprehensive: only computer vision tasks are included and only on toy datasets. I believe including recommendation system and NLP datasets will make the statement of using their new loss much more convincing.**
>
> In this paper, we aim to show that our loss function is compatible with existing DP-SGD improvement strategies [1] and [2], and allows us to establish new state-of-the-art tradeoffs between privacy and accuracy. Therefore, we used all three datasets that are used in these existing DP-SGD improvement strategies, please see [1] and [2].
> As pointed out in [1], there have not yet been any attempts to train more complex datasets such as ImageNet with DP-SGD, due to the cost of computing per-sample gradients and low accuracy of DP-SGD training. Like the reviewer, we hope that future work will scale DP-SGD to larger datasets but we think that our results are already valuable as is because CIFAR10 remains a challenging dataset for DP-SGD.
>
>
> [1] Florian Tramer and Dan Boneh. Differentially private learning needs better features (or much more data). ICLR, 2021.
> [2] Nicolas Papernot, Abhradeep Thakurta, Shuang Song, Steve Chien, and Ulfar Erlingsson. Tempered Sigmoid activations for deep learning with differential privacy. AAAI, 2021.
>
>
>
> > **Secondly, the empirical improvement is not significant. The seemly most significant improvement (as is the one presented in the abstract) is 4% on CIFAR10. Given that non-private model can easily get over 90% on CIFAR10, I believe the gap is not really closed by introducing the new loss.**
>
> We agree that there is still a big gap between the accuracy of non-private and private models. However, our 4% improvement is considered as a good accuracy improvement in the DP-SGD literature (as a reference please see the accuracy improvements of recent DP-SGD strategies in Table 2 of [1] and Table 4 of [2].)
> We also note that it may not be possible to “bridge the gap” between private and non-private learning because non-private learning is able to rely on overfitting and memorization, if necessary, to improve performance - whereas private learning cannot.
>
> [1] Wenxiao Wang, Tianhao Wang, Lun Wang, Nanqing Luo, Pan Zhou, Dawn Song, and Ruoxi Jia. DPlis: Boosting utility of differentially private deep learning via randomized smoothing. PETS, 2021.
> [2] Nicolas Papernot, Abhradeep Thakurta, Shuang Song, Steve Chien, and Ulfar Erlingsson. Tempered Sigmoid activations for deep learning with differential privacy. AAAI, 2021.
>
>
>
> > **Lastly, the lack of theory may make people wonder, does this improvement really hold for general DP training? At most we can say on specific datasets and specific models, using the new loss is beneficial. Also all three components of the new loss already exist in regular training, so the current approach seems a simple combination without sufficient justification.**
>
> We showed the benefit of deploying our proposed loss function in improving existing DP-SGD strategies by using all of the datasets and models previously used to benchmark the state-of-the-art approaches to DP-SGD. Our evaluation showed that our approach establishes new state-of-the-art tradeoffs between privacy and accuracy. On this note, we would like to refer to a comment from reviewer UK9m:  “the experiments are quite comprehensive uncovering the importance of choosing loss functions”. We also provided some justification in terms of information loss, convergence rate, generalisation and smoothness in Section 3.2. Finally, we would like to highlight that we conducted an ablation study in Section 4.3 to show all the components of our new loss are required for improving privacy-accuracy tradeoffs in deep learning with DP, so combining them was not trivial in itself.
> We are happy to expand any of these discussions if the reviewer sees it fit.

---

### Official Review · Reviewer_WSQr · 2021-11-01

**Correctness:** 3
**Technical Novelty And Significance:** 3
**Empirical Novelty And Significance:** 3
**Recommendation:** 5
**Confidence:** 4

**Main Review:**

Strength:

1.	Improving the utility of private deep learning from a loss design perspective is novel to me. This may be also valuable to the community.
2.	The model obtained by optimizing the proposed loss achieves the SOTA result.

Weakness:

The main disadvantage of this article is that it does not explain clearly the motivation for designing the loss and why the proposed loss can work. Although the author tried to analyze the mechanism of loss in section 3.2, these analyses were not convincing enough and did not clearly explain why the loss can work. Specifically,
1. The author indicated "Our loss limits information loss from clipping" through Fig. 1, but did not explain why the proposed loss can prevent logit values and weights from exploding.
2. The author claimed "Our loss yields faster convergence", but did not provide proof.
3. The author mentioned "Preventing logits from exploding may also help recover the generalization boost of batch normalization", which can only be used as a guess based on observations. A rigorous theoretical analysis may be required to strengthen the claim.
4. It seems that Figure 2 doesn't serve as an indication that the proposed loss function works because it enhances smoothness, because according to Fig 2, the smoothness of the network trained by CE loss is getting better and better, while the smoothness of the proposed loss training model is It's getting worse.

Another question: Are the parameters robust to different hyper-parameters settings in the loss?


**Summary Of The Paper:**

This paper attempts to improve the accuracy of the DP-SGD training from the perspective of loss function design. The authors propose a loss composed of SSE loss, focal loss, and L2 regularization penalty. Experiments are conducted to demonstrate the effectiveness of the
proposed loss.


**Summary Of The Review:**

None

---

> ### Author Response · Authors · 2021-11-16
> **Authors’ reply to Reviewer WSQr**
>
> We thank the reviewer for the comments, finding our paper novel and valuable for the community. The reviewer’s comments are in bold font, and the authors’ reply is in regular font.
>
>
> >_**Although the author tried to analyze the mechanism of loss in section 3.2, these analyses were not convincing enough and did not clearly explain why the loss can work. Specifically,**_
>
> We would like to emphasise that this is the first work that designs a loss tailored to DP-SGD training to recover some of the negative impact of DP-SGD on the performance. In the text below we split comments and respond to them inline.
>
> >_**The author indicated "Our loss limits information loss from clipping" through Fig. 1, but did not explain why the proposed loss can prevent logit values and weights from exploding.**_
>
> Our regularisation penalty on the intermediate pre-activations (see Equation 5) and Sum Squared Error between logit values and one-hot vector labels (see Equation 7) on the logit values both prevent logit values and weights from exploding. We will make this point more clear by changing\
> “In particular, we prevent logit values and weights from exploding, which decreases the magnitude and variance of per-example gradients (see Equation 3).”\
> to\
> “In particular, $\mathcal{L}_\text{Reg}$ on the intermediate pre-activations and $\mathcal{L}_\text{SSE}$ on the logit values both prevent logit values and weights from exploding, which decreases the magnitude and variance of per-example gradients (see Equation 3).“\
> in Section 3.2 of our manuscript. We also would like to note that we already mentioned about preventing weights from exploding in our manuscript before Equation 7 as\
> “To further decrease the gradient magnitudes and prevent the explosion of intermediate weights, we impose a regularisation penalty on the intermediate pre-activations as:”.
>
> >_**The author claimed "Our loss yields faster convergence", but did not provide proof.**_
>
> The fast convergence of our loss is because we replace the cross-entropy loss with the Sum Squared Error. Regarding the proof, we refer to [1] which proves that the Sum Squared Error converges exponentially while cross-entropy loss only converges at O(1/t).
>
> [1] Zeyuan Allen-Zhu, Yuanzhi Li, and Zhao Song. A convergence theory for deep learning via over- parameterization. ICML, 2019.
>
> >_**The author mentioned "Preventing logits from exploding may also help recover the generalization boost of batch normalization", which can only be used as a guess based on observations. A rigorous theoretical analysis may be required to strengthen the claim.**_
>
> Thank you for the suggestion. We agree that a rigorous theoretical analysis would allow us to strengthen the claim. However, it is not possible to integrate batch normalization in DP-SGD (analytically or experimentally) because it violates the per-example computation of gradients. This is why we instead made an argument similar to [1], which shows that preventing logits from exploding may also help recover the generalisation boost of batch normalisation. As our Sum Squared Error prevents the logits from exploding, we thus write “our proposed $L_SSE$ penalty may also be beneficial to learning with DP-SGD because it achieves a similar effect to batch normalisation” in Section 3.2.\
>
> [1] Yann Dauphin and Ekin Dogus Cubuk. Deconstructing the regularization of BatchNorm. ICLR, 2021.
>
> >_**It seems that Figure 2 doesn't serve as an indication that the proposed loss function works because it enhances smoothness, because according to Fig 2, the smoothness of the network trained by CE loss is getting better and better, while the smoothness of the proposed loss training model is It's getting worse.**_
>
> Empirically, in Figure 3 and Figure 4, we observe that smoothness of the loss function at the beginning of training with DP-SGD is important for improving the convergence rate and performance while weights remain close to their random initialization. In addition to this, the variance of singular values in cross-entropy is higher than the variance of singular values in our loss function.
>
> (*continued)

---

> > ### Author Response · Authors · 2021-11-18
> > **Authors’ reply to Reviewer WSQr**
> >
> > >**Another question: Are the parameters robust to different hyper-parameters settings in the loss?**
> >
> > We would like to thank the reviewer for this question. We performed experiments on all three datasets by varying the value of hyperparameter values of our loss function. Results show that performance has low sensitivity to these hyperparameter values. We added the results and the description of results in Appendix B of our manuscript as:\
> > ''However, we evaluate the performance of our loss function for each hyperparameter value using all three datasets in Figure 10. We can observe that the impact of hyperparameter values on the performance of our loss function is negligible. Therefore, the improvement of our loss function is not sensitive to the hyperparameter values.''

---

### Official Review · Reviewer_VgSd · 2021-11-01

**Correctness:** 2
**Technical Novelty And Significance:** 2
**Empirical Novelty And Significance:** 3
**Recommendation:** 5
**Confidence:** 5

**Main Review:**

There are several weak points of this paper.

The paper uses the theoretical evidence to argue that summed squared error is good for fast convergence. We have to admit that the convergence result in Allen-Zhu et al. 2019 has strict conditions: NTK regime, ReLU and square loss, which may not hold in the practical experiments. Even if these results hold, the fastness of exponential convergence is more evident for the final stage of convergence. However, here it is used to argue for the initial stage's fastness, which is not persuasive. In practice, the convergence rate is more related with the choices of the optimizers SGD or Adam, different learning rates. rather than different losses. The paper does not have good comparison with these choices.

The regularization penalty is quite similar to the function of the weight decay. How does it compare with normal weight decay? It is supposed that DP-SGD naïve application with weight decay can achieve better results than 59% on CIFAR10.

Most importantly, CIFAR10 may not be a good dataset for benchmarking DP algorithms. For one obvious reason, it is not a sensitive dataset from every aspects. It is a too small dataset, with each class 5000 samples so that the DP algorithm may not perform well as expected unless using pretrained model. Benchmarking DP algorithms on CIFAR10 may lead to over-optimized models and/or losses that are not be able to generalize to other real privacy-sensitive scenarios, i.e. language models.


**Summary Of The Paper:**

The paper proposes a tailored loss for DP-SGD, which includes summed squared error, the focal loss, and a regularization penalty. The summed squared error is for fast convergence at initial stage. The focal loss is used for identify hard samples. The regularization penalty is used for reducing the gradient/weight norm and avoid explosion. For each component of the tailored loss, the paper has empirical/theoretical evidence to argue the necessity. It is a good try to improve the performance of DP-SGD from amending the loss.

**Summary Of The Review:**

Good intuitions about DP algorithm but the results and arguments are not convincing. Overall, the review would like to give a weak reject to this paper.

---

> ### Author Response · Authors · 2021-11-16
> **Authors’ reply to Reviewer VgSd**
>
> We thank the reviewer for the comments. In the text below we split comments and respond to them inline. The reviewer’s comments are in bold font, and the authors’ reply is in regular font.
>
> >_**The paper uses the theoretical evidence to argue that summed squared error is good for fast convergence. We have to admit that the convergence result in Allen-Zhu et al. 2019 has strict conditions: NTK regime, ReLU and square loss, which may not hold in the practical experiments. Even if these results hold, the fastness of exponential convergence is more evident for the final stage of convergence. However, here it is used to argue for the initial stage's fastness, which is not persuasive.**_
>
> We did demonstrate that this fast convergence of sum squared error holds in our empirical evaluation. In Figure 9, we performed an ablation study showing that summed squared error converges faster than cross-entropy in the initial epochs of DP-SGD training.
> > **In practice, the convergence rate is more related with the choices of the optimizers SGD or Adam, different learning rates. rather than different losses. The paper does not have good comparison with these choices.**
>
> Optimiser:\
> Prior work found that the use of adaptive optimizers such as  Adam does not provide benefits for private learning. For instance,  SGD was compared to Adam in Table3 of [1] and the discussion in Section C.5 of [2]. This result is due to the fact that adaptive optimisers in DP-SGD carry noise from one gradient descent step to the next to adapt learning rates, therefore inadequately slowing down training [3].\
> Learning rate:\
> We agree that the choice of learning rate can affect the tradeoffs between privacy and accuracy in DP-SGD training. Therefore, we choose the best learning rate values reported in the hyperparameter search done by [1] and [2] to also have a fair comparison. Further tuning the learning rate would only strengthen and improve our results.
> We would like to mention that even the best learning rate values cannot compensate for the information that is discarded from the gradients due to per-example gradient clipping. This is because clipping is done at the granularity of individual training examples. Together with the noise injected, this biases learning and implies that DP-SGD takes a different optimisation path compared to SGD. Please see paragraphs 1 and 2 of Section 3 in our manuscript for more discussion.
> In conclusion, we believe this provides further evidence that designing a loss tailored to DP-SGD is a key overlooked aspect of learning when it comes to improving the utility of privacy-preserving learning.
>
> [1] Nicolas Papernot, Abhradeep Thakurta, Shuang Song, Steve Chien, and Ulfar Erlingsson. Tempered Sigmoid activations for deep learning with differential privacy. AAAI, 2021. (https://arxiv.org/pdf/2007.14191.pdf )\
> [2] Florian Tramer and Dan Boneh. Differentially private learning needs better features (or much more data). ICLR, 2021. (https://arxiv.org/pdf/2011.11660.pdf) \
> [3] Nicolas Papernot, Steve Chien, Shuang Song, Abhradeep Thakurta, and Ulfar Erlingsson. Making the shoe fit: Architectures, initializations, and tuning for learning with privacy, 2020a.  (https://openreview.net/pdf?id=rJg851rYwH)
> >**Most importantly, CIFAR10 may not be a good dataset for benchmarking DP algorithms. For one obvious reason, it is not a sensitive dataset from every aspects. It is a too small dataset, with each class 5000 samples so that the DP algorithm may not perform well as expected unless using pretrained model. Benchmarking DP algorithms on CIFAR10 may lead to over-optimized models and/or losses that are not be able to generalize to other real privacy-sensitive scenarios, i.e. language models.**
>
> In this paper, we aim to show that our loss function is compatible with existing DP-SGD improvement strategies [1] and [2], and allows us to establish new state-of-the-art tradeoffs between privacy and accuracy. Therefore, we used all three datasets that are used in these existing DP-SGD improvement strategies, please see [1] and [2].
> As pointed out in [1], there have not yet been any attempts to train more complex datasets such as ImageNet with DP-SGD, due to the cost of computing per-sample gradients and low accuracy of DP-SGD training. Like the reviewer, we hope that future work will scale DP-SGD to larger datasets but we think that our results are already valuable as is because CIFAR10 remains a challenging dataset for DP-SGD.
>
>
> [1] Florian Tramer and Dan Boneh. Differentially private learning needs better features (or much more data). ICLR, 2021.\
> [2] Nicolas Papernot, Abhradeep Thakurta, Shuang Song, Steve Chien, and Ulfar Erlingsson. Tempered Sigmoid activations for deep learning with differential privacy. AAAI, 2021.\
>
> (*continued)

---

> > ### Comment · Reviewer_VgSd · 2021-11-27
> > **reply to the authors' response**
> >
> > For the first point, it is almost impossible to rigorously argue which loss can achieve faster convergence in practice. The convergence rate is only an observation of a specific set of hyperparameter choices. It is also questionable for the "best performed" parameter. It is not clear what is the goal for hyper-parameter search: fastest initial convergence or the best final accuracy at a certain number of epochs?  As we know in non-private learning of CIFAR10, it may have different "best" hyperparameters if one want to achieve different goal.
> >
> > The response missed one comment of comparing with weight decay.
> >
> > Though I think the observation may be interested to the community but with very limited evidence to show the benefit of the proposal really generalizes to other important scenarios.

---

> > > ### Author Response · Authors · 2021-11-29
> > > **Authors’ reply to Reviewer VgSd**
> > >
> > > Thank you for your time reading our responses and posting new comments. In the text below we split comments and respond to them inline.
> > >
> > > > **For the first point, it is almost impossible to rigorously argue which loss can achieve faster convergence in practice. The convergence rate is only an observation of a specific set of hyperparameter choices.**
> > >
> > > We would like to highlight that we have two sets of hyperparameters: 1) DP-SGD related ones (Learning rate, Momentum, Batch size, Epochs, Noise scale and Clipping norm); and 2) loss function related ones. When using the summed squared error as the only loss function, we do not have any loss function related  hyperparameters to be tuned and for the DP-SGD related hyperparameters we use those that have been shown to perform best for cross-entropy in the existing works [1], [2]. Therefore, Figure 9 compares the convergence rate of  summed squared error versus  cross-entropy with the best values of DP-SGD related  hyperparameters for cross-entropy. Further tuning these DP-SGD related hyperparameters would only strengthen and improve our summed squared error results.
> > > In addition to this, as Reviewer UK9m suggested, we added the performance evaluation of our loss function for each hyperparameter value using all three datasets in Figure 10. We can observe that the impact of loss function related hyperparameter values on the performance of our loss function is negligible. Therefore, the improvement of our loss function is not sensitive to the hyperparameter values.
> > >
> > > > **It is also questionable for the "best performed" parameter. It is not clear what is the goal for hyper-parameter search: fastest initial convergence or the best final accuracy at a certain number of epochs? As we know in non-private learning of CIFAR10, it may have different "best" hyperparameters if one want to achieve different goal.**
> > >
> > > Existing works [1], [2] select the value of hyperparameters  by considering the best final accuracy for a fixed privacy budget. We followed this strategy too. However, Figure 3 and Figure 4 show that our final proposed loss function (SSE+Focal+Regulariser)  even achieves both goals simultaneously: the fastest initial convergence and the best final accuracy.

---

> > > > ### Author Response · Authors · 2021-11-29
> > > > **Authors’ reply to Reviewer VgSd**
> > > >
> > > > > **The response missed one comment of comparing with weight decay.**
> > > >
> > > > We would like to thank the reviewer for this suggestion, we have been working on its implementation in the Opacus DP library.  We performed experiments on all three datasets by replacing the pre-activation regulariser (PA\_Reg) with weight decay regulariser (WD\_Reg) in our final proposed loss function. As we cannot revise the manuscript at this phase anymore, we reported the results for CIFAR10 for 5 runs in the below table. Both pre-activation regulariser and weight decay regulariser help to improve the tradeoffs between privacy and accuracy of DP-SGD training. However, the improvement of pre-activation regulariser in both initial iterations and the maximum accuracy is higher than the one introduced by the weight decay regulariser.
> > > >
> > > > |$\epsilon$|SSE+Focal+PA\_Reg (min)|SSE+Focal+PA\_Reg (max)|SSE+Focal+PA\_Reg (med)|SSE+Focal+WD\_Reg (min)| SSE+Focal+WD\_Reg (max)|SSE+Focal+PWD\_Reg (med)|
> > > > |-------|----|--------|--------|--------|--------|-------|
> > > > |0.8263 |36.0900 |38.3700 |37.2200 |35.0900 |37.9700 |37.3200|
> > > > |0.9506 |42.3200 |43.7600 |43.4100 |41.7600 |43.4300 |42.3500|
> > > > |1.0661 |45.5800 |47.6900 |46.7000 |44.9600 |46.3300 |45.7400|
> > > > |1.1816 |48.4900 |50.6500 |50.1900 |46.1600 |48.0600 |47.9000|
> > > > |1.2924 |51.2300 |53.0000 |51.8200 |47.5500 |50.1600 |49.0200|
> > > > |1.3974 |53.1300 |54.1400 |53.7700 |48.1300 |50.5900 |50.5000|
> > > > |1.4963 |54.1600 |55.3100 |54.8900 |49.6000 |51.8800 |51.0100|
> > > > |1.5908 |55.6500 |56.5200 |56.1500 |50.4000 |52.7500 |51.9400|
> > > > |1.6789 |56.2300 |57.4500 |57.1000 |51.7700 |53.6400 |53.1400|
> > > > |1.7632 |57.7600 |58.3600 |57.9800 |52.9100 |54.8000 |54.3800|
> > > > |1.8436 |58.4000 |59.4200 |58.7800 |54.5400 |56.0500 |55.3400|
> > > > |1.9219 |58.8600 |59.9700 |59.2300 |55.5600 |56.9500 |56.3900|
> > > > |1.9948 |59.3900 |60.4300 |60.1700 |56.8700 |57.3900 |57.0800|
> > > > |2.0678 |59.4800 |60.6200 |60.1100 |57.6000 |58.1000 |58.0300|
> > > > |2.1390 |60.5400 |60.7700 |60.7400 |57.9800 |58.7400 |58.5000|
> > > > |2.2041 |60.4800 |61.7800 |60.7700 |58.7500 |59.5900 |59.0400|
> > > > |2.2691 |60.9100 |61.4500 |61.0800 |58.9300 |60.1700 |60.0400|
> > > > |2.3329 |60.7000 |62.1800 |61.2900 |59.1800 |60.8700 |60.2900|
> > > > |2.3951 |61.2000 |62.2300 |61.9700 |59.4400 |60.6500 |60.2800|
> > > > |2.4559 |60.0900 |62.4000 |61.1300 |59.6100 |61.2700 |60.8600|
> > > > |2.5153 |61.3600 |62.3300 |61.7800 |60.2500 |61.4400 |61.2600|
> > > > |2.5735 |61.6400 |62.4900 |61.6900 |60.7700 |61.3900 |61.1500|
> > > > |2.6305 |61.3700 |62.4600 |62.1800 |60.9400 |61.5800 |61.1000|
> > > > |2.6863 |61.6200 |62.8100 |62.3800 |61.4700 |62.1300 |61.6500|
> > > > |2.7412 |62.0700 |62.7500 |62.3700 |61.8300 |62.1500 |61.9200|
> > > > |2.7950 |62.0900 |62.8100 |62.1900 |61.4800 |62.2900 |61.8600|
> > > > |2.8480 |62.1200 |62.8100 |62.5500 |61.2100 |62.4700 |61.8300|
> > > > |2.9000 |61.8100 |63.0100 |62.6500 |61.4300 |62.1600 |61.8000|
> > > > |2.9512 |62.1400 |62.8400 |62.5900 |60.9300 |62.2700 |61.4800|
> > > > |3.0017 |62.3200 |63.1900 |62.6700 |61.1700 |62.4500 |61.6400|
> > > >
> > > >
> > > > We are happy to add the results of comparison with weight decay using all datasets in the manuscript.
> > > >
> > > >
> > > > > **Though I think the observation may be interested to the community but with very limited evidence to show the benefit of the proposal really generalizes to other important scenarios.**
> > > >
> > > > Thanks for considering our work interesting for the community. In this work, we showed that the benefit of our proposed loss function generalizes to both state-of-the-art DP-SGD improvement strategies [1], [2] using all of the datasets and models that they have used and which are common in the DP-SGD literature. We would also like to note that one of our main novelties is demonstrating the need for (and advantages of) designing loss functions tailored to DP-SGD to reduce the gap between the accuracy of private and non-private models, generally speaking.
> > > >
> > > >
> > > > [1] Florian Tramer and Dan Boneh. Differentially private learning needs better features (or much more data). ICLR, 2021.
> > > >
> > > > [2] Nicolas Papernot, Abhradeep Thakurta, Shuang Song, Steve Chien, and Ulfar Erlingsson. Tempered Sigmoid activations for deep learning with differential privacy. AAAI, 2021.

---

### Official Review · Reviewer_UK9m · 2021-11-03

**Correctness:** 3
**Technical Novelty And Significance:** 2
**Empirical Novelty And Significance:** 3
**Recommendation:** 6
**Confidence:** 3

**Main Review:**

Overall I think this is an interesting paper that explores the impact of loss function on the performance of models trained by DP-SGD. Also, the experiments are quite comprehensive uncovering the importance of choosing loss functions. Yet, there are still some aspects that could be improved.

1. Though the loss can improve upon cross-entropy, it introduces three extra hyperparameters to tune. It seems the paper only reported the best performance after hyperparameter searches. However, the sensitivity of performance with respect to these hyperparameters is not reported or discussed. It would be better if such results can be added.

2. The impact of loss function on model performance is an important and interesting topic. The loss function proposed by the paper consists of three components, all from existing literature. I wonder have the authors tried other components (e.g. using hinge loss of Huber loss to replace the sum of squared errors or using other penalties)?

-----------------------after rebuttal--------------------------------------

My concerns are mostly addressed. As pointed out by other reviewers, some intuitive arguments for motivating the losses are not well-supported and misleading (e.g. using Allen-Zhu et al. 2019 to argue faster convergence in practice mentioned by Reviewer VgSd). However, I believe this paper should be viewed as an empirical paper and I am satisfied with the authors' efforts in exploring the impact of loss function on the performance of DP-SGD. Thus, I would like to keep my score.


**Summary Of The Paper:**

The paper proposes a new loss function to improve the performance of neural network models trained by DP-SGD. The new loss function is a weighted average of the sum of squared error, the focal loss, and a penalty on the squared norm of the pre-activation output of different layers. The new loss achieves state-of-the-art accuracy on the CIFAR-10, FashionMNIST, and MNIST datasets. It is also shown that the new loss can reduce the bias of gradient clipping and encourage learning on hard examples.

**Summary Of The Review:**

Overall I think the paper is interesting and could be further improved if more results are added.

---

> ### Author Response · Authors · 2021-11-18
> **Authors’ reply to Reviewer UK9m**
>
> We thank the reviewer for the comments, and for highlighting that our paper is interesting and comprehensive in its experiments. In the following, we respond to your comments inline.
> > **Though the loss can improve upon cross-entropy, it introduces three extra hyperparameters to tune. It seems the paper only reported the best performance after hyperparameter searches. However, the sensitivity of performance with respect to these hyperparameters is not reported or discussed. It would be better if such results can be added.**
>
> We would like to thank the reviewer for this suggestion. We performed experiments on all three datasets by varying the value of hyperparameter values of our loss function. Results show that performance has low sensitivity to these hyperparameter values. We added the results and the description of results in Appendix B of our manuscript as:\
> ''However, we evaluate the performance of our loss function for each hyperparameter value using all three datasets in Figure 10. We can observe that the impact of hyperparameter values on the performance of our loss function is negligible. Therefore, the improvement of our loss function is not sensitive to the hyperparameter values.''
>
> > **The impact of loss function on model performance is an important and interesting topic. The loss function proposed by the paper consists of three components, all from existing literature. I wonder have the authors tried other components (e.g. using hinge loss of Huber loss to replace the sum of squared errors or using other penalties)?**
>
> Thank you for this suggestion. In Appendix F, we added an ablation study on the tradeoffs between privacy and accuracy of Sum Squared Error, Sum Absolute Error and Huber loss in DP-SGD training. Figure 14 shows that Sum Squared Error achieves the best tradeoff between accuracy and privacy in all three datasets among Sum Squared Error, Sum Absolute Error and Huber loss.

---

> > ### Author Response · Authors · 2021-11-29
> > **Authors’ reply to Reviewer UK9m**
> >
> > > **Using other penalties**.
> >
> > We would like to thank the reviewer for this suggestion, we have been working on the implementation of weight decay penalty in the Opacus DP library instead of our pre-activation penalty as suggested by Reviewer VgSd.  We performed experiments on all three datasets by replacing the pre-activation regulariser with weight decay regulariser in our final proposed loss function. As we cannot revise the manuscript at this phase anymore, we reported the results for CIFAR10 for 5 runs in the below table. Both pre-activation regulariser and weight decay regulariser help to improve the tradeoffs between privacy and accuracy of DP-SGD training. However, the improvement of pre-activation regulariser in both initial iterations and the maximum accuracy is higher than the one introduced by the weight decay regulariser.
> >
> > |$\epsilon$|SSE+Focal+PA\_Reg (min)|SSE+Focal+PA\_Reg (max)|SSE+Focal+PA\_Reg (med)|SSE+Focal+WD\_Reg (min)|SSE+Focal+WD\_Reg (max)| SSE+Focal+PWD\_Reg (med)|
> > |-------|--------|--------|--------|--------|--------|-------|
> > |0.8263 |36.0900 |38.3700 |37.2200 |35.0900 |37.9700 |37.3200|
> > |0.9506 |42.3200 |43.7600 |43.4100 |41.7600 |43.4300 |42.3500|
> > |1.0661 |45.5800 |47.6900 |46.7000 |44.9600 |46.3300 |45.7400|
> > |1.1816 |48.4900 |50.6500 |50.1900 |46.1600 |48.0600 |47.9000|
> > |1.2924 |51.2300 |53.0000 |51.8200 |47.5500 |50.1600 |49.0200|
> > |1.3974 |53.1300 |54.1400 |53.7700 |48.1300 |50.5900 |50.5000|
> > |1.4963 |54.1600 |55.3100 |54.8900 |49.6000 |51.8800 |51.0100|
> > |1.5908 |55.6500 |56.5200 |56.1500 |50.4000 |52.7500 |51.9400|
> > |1.6789 |56.2300 |57.4500 |57.1000 |51.7700 |53.6400 |53.1400|
> > |1.7632 |57.7600 |58.3600 |57.9800 |52.9100 |54.8000 |54.3800|
> > |1.8436 |58.4000 |59.4200 |58.7800 |54.5400 |56.0500 |55.3400|
> > |1.9219 |58.8600 |59.9700 |59.2300 |55.5600 |56.9500 |56.3900|
> > |1.9948 |59.3900 |60.4300 |60.1700 |56.8700 |57.3900 |57.0800|
> > |2.0678 |59.4800 |60.6200 |60.1100 |57.6000 |58.1000 |58.0300|
> > |2.1390 |60.5400 |60.7700 |60.7400 |57.9800 |58.7400 |58.5000|
> > |2.2041 |60.4800 |61.7800 |60.7700 |58.7500 |59.5900 |59.0400|
> > |2.2691 |60.9100 |61.4500 |61.0800 |58.9300 |60.1700 |60.0400|
> > |2.3329 |60.7000 |62.1800 |61.2900 |59.1800 |60.8700 |60.2900|
> > |2.3951 |61.2000 |62.2300 |61.9700 |59.4400 |60.6500 |60.2800|
> > |2.4559 |60.0900 |62.4000 |61.1300 |59.6100 |61.2700 |60.8600|
> > |2.5153 |61.3600 |62.3300 |61.7800 |60.2500 |61.4400 |61.2600|
> > |2.5735 |61.6400 |62.4900 |61.6900 |60.7700 |61.3900 |61.1500|
> > |2.6305 |61.3700 |62.4600 |62.1800 |60.9400 |61.5800 |61.1000|
> > |2.6863 |61.6200 |62.8100 |62.3800 |61.4700 |62.1300 |61.6500|
> > |2.7412 |62.0700 |62.7500 |62.3700 |61.8300 |62.1500 |61.9200|
> > |2.7950 |62.0900 |62.8100 |62.1900 |61.4800 |62.2900 |61.8600|
> > |2.8480 |62.1200 |62.8100 |62.5500 |61.2100 |62.4700 |61.8300|
> > |2.9000 |61.8100 |63.0100 |62.6500 |61.4300 |62.1600 |61.8000|
> > |2.9512 |62.1400 |62.8400 |62.5900 |60.9300 |62.2700 |61.4800|
> > |3.0017 |62.3200 |63.1900 |62.6700 |61.1700 |62.4500 |61.6400|
> >
> >
> > We are happy to add the results of comparison with weight decay using all datasets in the manuscript.

---

### Author Response · Authors · 2021-11-19
**General comment**

Many thanks dear reviewers for your time! We would be happy to answer any further questions you may have before the response period ends on Monday.

---

### Decision · Program_Chairs · 2022-01-20

**Decision:**

Reject

**Comment:**

The reviewers all seemed to agree that the investigation of other losses is an interesting direction of study, and acknowledged there was some empirical performance improvement for standard computer vision tasks. However, they felt the justification of the specific form of loss was a bit shaky and heuristic, and were furthermore unconvinced by results exclusively for image classification (one reviewer was unmoved by the magnitude of improvement). This was a borderline decision, but we hope the authors refine and resubmit their work as this is an interesting but underexplored direction within DPML.

As one recent related work which investigates the effect of other architecture differences in the DP setting, the authors may be interested in https://arxiv.org/abs/2110.08557.